# Magnetic Hydroxyapatite Nanoparticles in Regenerative Medicine and Nanomedicine

**DOI:** 10.3390/ijms25052809

**Published:** 2024-02-28

**Authors:** Hina Inam, Simone Sprio, Marta Tavoni, Zahid Abbas, Federico Pupilli, Anna Tampieri

**Affiliations:** 1Institute of Science, Technology and Sustainability for Ceramics (ISSMC), National Research Council of Italy (CNR), 48018 Faenza, Italy; hina.inam@issmc.cnr.it (H.I.); marta.tavoni@issmc.cnr.it (M.T.); zahid.abbas@issmc.cnr.it (Z.A.); federico.pupilli@issmc.cnr.it (F.P.); 2Department of Material Science and Technology, University of Parma, 43121 Parma, Italy; 3Department of Chemistry “Giacomo Ciamician”, University of Bologna, 40126 Bologna, Italy; 4Department of Chemical Sciences, University of Padova, 35122 Padova, Italy

**Keywords:** magnetic hydroxyapatite, synthesis methods, drug delivery, regenerative medicine, nano medicine

## Abstract

This review focuses on the latest advancements in magnetic hydroxyapatite (mHA) nanoparticles and their potential applications in nanomedicine and regenerative medicine. mHA nanoparticles have gained significant interest over the last few years for their great potential, offering advanced multi-therapeutic strategies because of their biocompatibility, bioactivity, and unique physicochemical features, enabling on-demand activation and control. The most relevant synthetic methods to obtain magnetic apatite-based materials, either in the form of iron-doped HA nanoparticles showing intrinsic magnetic properties or composite/hybrid compounds between HA and superparamagnetic metal oxide nanoparticles, are described as highlighting structure–property correlations. Following this, this review discusses the application of various magnetic hydroxyapatite nanomaterials in bone regeneration and nanomedicine. Finally, novel perspectives are investigated with respect to the ability of mHA nanoparticles to improve nanocarriers with homogeneous structures to promote multifunctional biological applications, such as cell stimulation and instruction, antimicrobial activity, and drug release with on-demand triggering.

## 1. Introduction

Magnetic nanomaterials (MNPs) have attained significant interest in a wide range of applications, including imaging, catalysis, nanofluids, colloidal photonic crystals, data storage, optical filters, defect sensing, and environmental remediation [1,2,3]. Moreover, MNPs also have a potential use in biomedical applications, such as regenerative medicine and nano-medicine, leveraging on the smart functionalities enabled by magnetic properties—though such properties have to be associated with low toxicity and appropriate physicochemical properties to enable use in vivo [4,5,6,7,8]. Currently, the most widely used MNPs are based on iron, nickel, manganese, cobalt, gadolinium, and their compounds [9,10,11]. Particularly, iron is categorized as a trace element within the human body and has the ability to undergo metabolism [12]. Iron based MNPs for use in medicine are commonly referred to paramagnetic or superparamagnetic compounds, therefore exhibiting magnetic properties only when an external magnetic field is applied, and refer to two different general categories: (i) iron-oxide-based magnetic nanoparticles (SPIONs such as maghemite and magnetite) and (ii) iron-doped biocompatible and bioactive nanoparticles such as iron-doped hydroxyapatite [13,14,15,16]. The oxides of iron, together with those of gadolinium and manganese have also gained considerable attention in biomedical imaging (MRI), due to their relative low toxicity and feasibility of tailoring magnetic behavior [17,18,19,20]. These fascinating properties enable their use in the controlled delivery of drugs and bioactive molecules, MRI, cell separation, molecular labeling, and tissue growth [21,22,23,24]. A wide range of materials such as ceramics, metals, polymers, and their composites, are utilized for developing MNPs for various biomedical applications [25,26,27,28]. Recently, researchers have also paid significant attention to developing magnetic nanomaterial composites (MNCs) by combining two or more components to produce integrated nano-systems enabling bio-relevant multifunctional properties [29,30,31,32,33,34,35,36,37,38,39,40,41,42,43,44,45,46,47,48]. From a material science perspective, the handling and synthesis of MNCs may be difficult due to their low uniformity, the presence of impurities, and a high tendency to agglomerate [49,50]. Therefore, although composites offer tailored properties through materials of choice, these benefits are often overwhelmed by challenges in synthesis consistency, handling, and biocompatibility issues.

MNPs, such as SPIONs, have attracted a lot of attention in the field of nano-medicine due to higher saturation magnetization and magnetic susceptibility [26,51,52,53,54]. Regarding their impact on cells, many studies indicate that iron oxide nanoparticles exhibit a fair biocompatibility and generally remain inert towards cells under normal conditions, even though there are some potential adverse effects such as the generation of reactive oxygen species (ROS) in cells through the Fenton reaction [55] and the triggering of inflammatory responses under certain conditions, particularly when subjected to an external magnetic field [56,57,58,59,60,61]. To prevent or limit this drawback, SPIONs are often developed with a thin external coating to limit cytotoxicity effects [62,63]. Nevertheless, the accumulation in vivo of SPIONs in some organs, such as the kidney and liver, has been documented as a long-term cytotoxicity problem. Such drawbacks raise complications in regulatory evaluations [64,65,66,67], so that most magnetic-based materials and nanocomposites may not be directly applicable.

To circumvent these issues, material scientists are increasingly pushed to develop new solutions based on biocompatible and bioactive materials, where magnetic properties are conferred by doping with ions such as Fe, Ni, and Co, with the purpose to combine the material bio-degradability and the ability to be magnetized minimizing the cytotoxicity of nanomaterials like SPIONS [68,69,70,71].

The most popular inorganic material in regenerative medicine is hydroxyapatite (HA, general formula: Ca_5_(PO_4_)_3_OH), a calcium phosphate mimicking the composition of human mineralized tissues, so that it shows excellent biocompatibility, bioactivity, osteoinductivity, biodegradability, and non-immunogenic behavior [72,73,74,75,76,77,78]. For this reason, HA is since decades considered as the reference material for application in bone tissue engineering, more recently also in soft tissue regeneration with antibacterial capability, and also explored as vehicles for drug delivery, thanks to its ability to link a variety of drugs and biological molecules [79,80,81,82,83,84,85]. A relevant feature of nanocrystalline HA is the ability to host a variety of foreign ions such as Mg^2+^, Sr^2+^, Fe^2+^/Fe^3+^, CO_3_^2−^, Zn^2+^, Cu^2+^, and F^−^ in a partial substitution of Ca^2+^, PO_4_^3−^ and OH^−^. Such ionic doping/substitutions alter the thermodynamic stability of the HA crystal as well as its bulk and surface properties while retaining its lattice structure. Several studies have explored the synthesis of variously doped HA nanoparticles to obtain materials with enhanced biologic abilities [86,87,88,89,90,91,92,93,94,95,96]. On the other hand, the doping of HA with magnetic ions (such as Fe) has gained considerable attention as a switching system to remotely control and tune the regenerative process, for its ability to become superparamagnetic under the application of static magnetic field, as well as acting as a contrast agent for magnetic resonance imaging (MRI) [97] and a heat mediator for hyperthermia-based anti-cancer therapies when an alternating magnetic field is applied [98,99]. The type and concentration of iron ions inside the HA lattice play a crucial role in influencing the properties of magnetic HA [82]. For instance, it has been reported that the crystallite size and crystallinity of HA particles are reduced by increasing the Fe ions concentration, where the magnetic behavior shifted from diamagnetic to superparamagnetic [74,75,76,77,78,79,80,81,82,83].

Most of the traditional approaches targeting the magnetic HA phase result in the development of MNCs based on hydroxyapatite and iron oxides (magnetite and maghemite). More recently, to contrast the possible adverse reactions elicited by the presence of iron oxides, intrinsically magnetic iron-doped hydroxyapatite (FeHA) has been developed. Its magnetic properties are yielded by lattice mismatches induced by Fe^2+^/Fe^3+^ ions doping and specific positioning in the HA lattice and/or surface [100]. This makes FeHA biocompatible and bioactive, showing additional superparamagnetic properties and, thus, being promising for multiple bio-functionalities under applied low magnetic fields [100,101,102]. Furthermore, it was reported that the magnetization ability and hyperthermia properties of FeHA are comparable, or even superior, to those of magnetite or maghemite particles despite the iron content results being much lower [103,104,105].

In this review, recent advances in the development and use of apatitic magnetic nanoparticles (mHA) and related composites as promising candidates for biomedical applications at the frontier are discussed. The review will recapitulate the state-of-the-art related to the development of apatite-based magnetic nanoparticles and composites by various synthesis methods and, on the other hand, will dedicate a section to the description of single-phase Fe-doped apatites. Furthermore, relevant advances in biomedical applications of MNC and MNP, considering critical needs and challenges in regenerative medicine and nanomedicine, are discussed.

## 2. Methods of Magnetic Hydroxyapatite Synthesis

Various synthesis methods have been investigated for mHA synthesis, broadly depicted in Figure 1. These methods specifically affect relevant the physicochemical features of the obtained material, such as size, shape, and crystal structure, in turn related to specific bio-relevant magnetic properties. In particular, we will describe the achievement of: (a) HA-based MNCs, (b) HA doped with ions, providing intrinsic magnetic properties, making specific distinctions between these two types of materials for each synthesis method described.

### 2.1. Chemical Precipitation

The chemical precipitation approach is a simple method to generate HA-based MNP and related composites (MNCs) and to control the nanomaterials’ composition and size, making it a valuable tool for producing homogeneous particles with the possibility to tailor their surface features. Homogeneous precipitation and co-precipitation are the two primary chemical methods used for the fabrication of mHA. The homogeneous precipitation approach has many benefits, including the production of uniformly sized HA MNP, an easy and cost-effective process, whereas co-precipitation produces particles with different morphologies and high purity [106,107,108,109]. The chemical precipitation technique typically undergoes several processes. Initially, reagents containing calcium and phosphate are mixed; examples of which are mixtures of calcium hydroxide or calcium nitrate providing Ca^2+^, and orthophosphoric acid or diammonium hydrogen phosphate providing PO_4_^3−^, prepared according to the Ca/P ratio of 1.67 which is the value for stoichiometric hydroxyapatite. Subsequently, the pH of this mixture is adjusted; often an alkaline pH is adjusted in the region 7–12 where HA is thermodynamically stable among other calcium phosphates, while the reaction temperature can vary within the range of room temperature up to the boiling point of water. Incorporating metal cations (M⁺ = Zn^2^⁺, Fe^3^⁺, Ni^2^⁺, Mn^2^⁺, Co^2^⁺, Cu^2^⁺) in the structure of hydroxyapatite is carried out by dissolving soluble salts, such as nitrates or chlorides, in the Ca-containing solution. Ion-doped hydroxyapatites have been widely described in past literature as biomaterials with largely improved physical, chemical, and biological properties [110,111,112,113,114,115,116,117,118].

For the synthesis of a magnetic HA-based composite, iron oxides (e.g., Fe_3_O_4_) are introduced into the solution during their synthesis or their formation, and are induced during the crystallization of hydroxyapatite [107]. These nanoparticles become incorporated into the growing hydroxyapatite particles, resulting in a magnetic HA-based product.

After the precipitation process, the solution is stirred for aging, and then, the resulting precipitates are washed through centrifuge, dried and grinded into powder form. Bioactive hydroxyapatite powders are often produced by this method because of its low cost, simple functionality, small product particles, and excellent purity. However, the HAP particles produced by the precipitation technique exhibit poor homogeneity and are prone to agglomeration, potentially resulting in the formation of calcium-deficient apatite with lattice imperfections and a Ca/P ratio less than 1.67. Hence, the control requirements on the preparation process conditions are quite challenging. Therefore, it is crucial to strictly regulate the reaction conditions to ensure a complete reaction at an optimal temperature, pH, and reaction time [119,120,121].

#### 2.1.1. HA-Based MNCs Obtained by Chemical Precipitation

HA-based MNCs have great potential for use in the biomedical field, for example, as carriers for magnetically controlled drug delivery and materials for bone tissue engineering. After synthesis by a co-precipitation method to sinter iron oxides (Fe_3_O_4_ and γ-Fe_2_O_3_), the obtained agglomerated composite made of HA and iron oxides exhibited very low ferromagnetic characteristics, which was demonstrated by a saturation field of around 30 kOe. The synthesized composite is promising as a vehicle for targeted drug delivery due to its combination of biocompatible and ferromagnetic components. Weak ferromagnetic properties prevent particles from clumping together, ensuring they remain uniformly dispersed out in the desired location, which promotes cellular absorption and drugs delivery. Bioactive nanosystems based on the synthesized magnetic composites (HA-Fe_x_O_y_) with adsorbed biologically active substances have been developed for potential in vivo applications, particularly in the field of bone disease treatment [122].

Magnetite-HA nanocomposites produced by single-step wet chemical method include a magnetite phase that is well-crystallized and has a greater magnetic susceptibility and superparamagnetic characteristics, as related to the lower crystallite size of the samples. Co-precipitation involves separate precipitation of calcium and phosphate ions, while the more rapid single-step method incorporates the addition of precursors simultaneously. The samples produced by a single-step approach were found to have smaller magnetite crystallites with a higher surface area-to-volume ratio, allowing for a strong interaction between the material and the body’s cells, which can aid in tissue regeneration and repair by promoting cell adhesion and proliferation [123].

In another work, HA-based nanomaterials, including magnetic iron oxides, were produced by a wet chemical precipitation method under alkaline conditions. When Fe_3_O_4_ is added to hydroxyapatite during the coprecipitation process, the resulting HA/Fe_3_O_4_ composite has a slightly higher surface area and porosity than pure HA, primarily due to the presence of iron oxide. The incorporation of Fe_3_O_4_ into HA during precipitation increases the surface area and porosity of the resulting composite, primarily due to the inhibition of HA agglomeration and the increase of porosity, nucleation, aggregation, and particle growth that typically occur during wet precipitation. This improved structure makes the composite promising for various biomedical applications, such as tissue engineering and targeted drug delivery, where magnetic particles can be directed to specific parts of the body [124].

#### 2.1.2. HA Doped with Ions Giving Intrinsic Magnetic Properties Obtained by Chemical Precipitation

Fe^2+^/Fe^3+^-doped hydroxyapatite nanoparticles (FeHA) were synthesized using a neutralization method by the simultaneous addition of iron chlorides in a reaction vessel containing calcium hydroxide, whereas ortophosphoric acid was dropped at a temperature < 40 °C to prevent the crystallization of iron oxides [83]. The low temperature and the maintenance of alkaline conditions favored the entering of Fe ions into the HA lattice as confirmed by XRD and XAS structural analyses, with minimal formation of maghemite (<2%) on the HA surface. The FeHA showed a superparamagnetic-like behavior typical of single-domain magnetic nanoparticles, high magnetization saturation (4.0–4.2 A m^2^ kg^−1^) and hyperthermia effect, higher than that obtained with a mixture of apatite and maghemite nanoparticles. A subsequent study confirmed that the magnetization properties of FeHA could not be ascribed to the very low extent of maghemite formed during the synthesis but rather to the interplay of phenomena activated by Fe^2+^ and Fe^3+^ ions located in the nanocrystalline HA lattice, and by Fe^3+^ ions present in its amorphous superficial layer, synergistically yielding intrinsic superparamagnetic properties [15]. The absence of iron oxides made FeHA biocompatible for osteoblasts and bioresorbable, as typical of ion-doped HA.

Ullah et al. (2020) produced Fe/Sr co-doped HA with versatile particle sizes ranging from 140 to 205 nm by a sonication-assisted chemical aqueous precipitation approach, using calcium nitrate tetrahydrate (Ca(NO_3_)_2_·4H_2_O), strontium nitrate (Sr(NO_3_)_2_), ferric chloride hexahydrate (FeCl_3_·6H_2_O), diammonium hydrogen phosphate ((NH_4_)_2_HPO_4_), and ammonia (NH_3_) solution. Pore diameter and surface area of nanoparticles were about between 13 and 19 nm and 186 m^2^/g, respectively. The zeta potential value of the particles was significantly negative at pH 7. The results showed that the addition of Fe^3+^ to HA reduced thermal phase stability compared to both pure HA and Sr-HA. However, alkaline phosphatase (ALP) activity and calcium deposition were both enhanced by a synergistic impact from co-doping with Fe^3+^ and Sr^2+^ [125].

In another study, the same research team reported the biological activities of Fe/Sr-modified HA bioceramic composites with a particle size less than 0.8 μm, which were synthesized by previous precipitation approach [125]. The biological efficacy of these composites was evaluated using different assays, such as the acid citrate dextrose (ACD) test for in vitro blood compatibility, the alkaline phosphatase test for osteoblastic cell differentiation assessment, and an in vitro drug loading and release analysis. The synergistic effect of foreign ions (Sr and Fe) incorporated into the HA lattice and sintering methods resulted in active surface functional groups that enhanced drug loading and prolonged release patterns of the drugs 5-fluorouracil and amoxicillin at the pH level of a physiological medium. The findings of these tests have proven the biocompatibility of the composites, thus ensuring the potential biomedical application of these nanocomposites [126].

### 2.2. Mechanochemical Method

The mechanochemical synthesis method is a dry/solvent-free or liquid-assisted technique used to produce different nanomaterials by direct chemical reaction of compounds triggered by mechanical phenomena. This approach offers several advantages, including its environmentally friendly nature, low cost, and straightforward operational procedures [127]. The mechanochemical method is also utilized for the preparation of many heterogeneous nanoparticles, and key factors such as precursors, milling speed, time, and the choice of milling media play a crucial role in this process. This method can be categorized as either dry or wet, depending on whether a solvent is involved during milling. In the dry mechanochemical method, no solvent is used, whereas the wet mechanochemical method involves the addition of a small amount of solvent [128,129]. Although the mechanochemical method has proven to be valuable for synthesizing magnetic HA nanopowders and achieving relatively high crystallinity, its application in magnetic HA production is constrained by certain limitations. These limitations include the formation of non-homogeneous structures, particle aggregation, and the significant time required for the process [130].

#### 2.2.1. HA-Based MNCs Obtained by Mechanochemical Method

Superparamagnetic and biocompatible HA/Fe_3_O_4_ nanocomposites with a ratio of 1.5:1 (*w*/*w*) were synthesized by the wet mechanochemical approach (300 rpm for 5 h). Synthesized particles had a spherical shape and varied in size from 100 nm to 350 nm. The synthesized composites, using an efficient mechanochemical technique, were shown to be superparamagnetic and biocompatible, making them ideal for bone cancer treatment and other biomedical applications [131].

In another work, a simple mechanochemical technique was presented, which involves the rapid sequential preparation of superparamagnetic Fe_3_O_4_ nanoparticles and submicron-sized HA particles at room temperature, followed by the efficient incorporation of the Fe_3_O_4_ nanoparticles into the HA matrix by the milling process. Fe_3_O_4_ nanoparticles were homogeneously dispersed in the produced Fe_3_O_4_/HA composites without the formation of any significant agglomerates, even in the absence of anti-agglomeration agents. When the Fe_3_O_4_/HA composites were subjected to an alternating magnetic field, the heat generated increased with the increasing magnetite concentration. At a magnetite concentration of 30 wt%, the temperature increased by over 20 K within only 50 s. Super-paramagnetic behavior with magnetization of 78 emu/g was observed in highly crystalline Fe_3_O_4_ particles with an average diameter of 16 nm. It was also found that the HA powder produced by this process was low crystalline B-type carbonate HA, making it an ideal bone-substitute material. Thus, this synthesis approach demonstrates high efficiency in producing Fe_3_O_4_/HA composites, which hold potential for application in hyperthermia therapy targeting malignant bone tumors [132].

#### 2.2.2. HA Doped with Ions Giving Intrinsic Magnetic Properties Obtained by Mechanochemical Method

The mechanochemical technique with a dry approach was used to produce iron-substituted HA. When iron cations are introduced as a dopant in the mechanochemical synthesis of HA, a structural modification occurs where the dopant replaces the position of the calcium cation. This substitution leads to a reduction in the lattice parameters of HA. Additionally, it is also found that, during the mechanochemical synthesis process, under specific conditions, there is a possibility of simultaneous partial substitution of calcium cations with iron cations and phosphate groups with the carbonate group [133]. The obtained iron ion-doped hydroxyapatite (Fe-HA) has magnetic properties that have significance in the fields of biology and medicine. Magnetic hydroxyapatite synthesized via the mechanochemical route has been found to be used as heating mediators in hyperthermic cancer treatment [134,135].

Iron- and silicon-substituted HAs were synthesized by the mechanochemical method where iron cations and silicate groups replaced calcium and phosphate ions, respectively. Amorphous HA was produced when the substitution level was increased to 2.0, whereas single-phase HA was produced with a small amount of water up to a 1.5 substitution level. It is reported that HA with substitution levels below 0.5 partially decomposed at 800 °C, but HA with larger substitution levels decomposed at lower temperatures. At temperatures over 700 °C, the HA structure may undergo degradation, leading to a reduction in its crystallinity and mechanical properties. The production of iron oxide at higher temperatures indicates the release of iron cations from the apatite phase. The limit for simultaneous substitution of iron and silicon ions in HA has been reported to be less than 1. The findings of this study showed that the synthesized materials have potential to be used in bio-applications like medical devices due to its enhanced biological properties as compared to pure HA; however, optimal heat treatment temperature should not exceed 700 °C to maintain material stability for biomedical applications [136].

### 2.3. Emulsion Synthesis Method

The emulsion synthesis method involves the formation of micro-emulsions, which possess distinct characteristics such as controlled size, specific morphology, and minimal aggregation, leading to the production of well-defined nanoparticles with tailored properties. Micro-emulsions are formed by combining two immiscible phases, typically water and oil, and their stability is achieved with the use of surfactants. Under optimal conditions, such as low-pressure and low-temperature, this approach enables the production of magnetic hydroxyapatite microbeads, core–shell structures, and microspheres with little material aggregation.

Hollow superparamagnetic iron-substituted hydroxyapatite (FeHA) nano-microspheres of 2 μm to 500 nm with high biocompatibility were synthesized by using emulsifiers like hybrid polymeric materials such poly L-lactic acid and CH_2_Cl_2_ [101]. FeHA was obtained by a method reported by Tampieri et al. [137]. Several hybrid magnetic composites within a polymeric matrix were synthesized, controlling chemico-physical properties, such as their size, surface charge, and magnetization, by varying the amount of FeHA (from 1 to 30 wt%). The synthesized magnetic composite with the largest amount of inorganic phase showed the smallest size, as well as the highest surface charge and magnetization. Furthermore, higher amounts of FeHA were correlated with higher cell proliferation rate. The potential to control the FeHA content of composites is promising to improve their outstanding biological performance to respond to specific needs in medical applications [138].

### 2.4. Hydrothermal Method

In general, hydrothermal processes include a chemical reaction in an aqueous solution that occurs at elevated temperatures and pressures and are widely used to produce crystalline nanoparticles. During the fabrication process, the reactant mixture is introduced into an autoclave that operates under high pressure and temperature. Studies have shown that the regulation of hydrothermal temperature, process time, and concentration of reactants may effectively control the shape and size of the final product. An interesting feature of this approach includes the use of either an autoclave or a microwave to achieve high levels of pressure and temperature, which facilitates the formation of stoichiometric compounds with the desired crystalline structure with controlled morphology such as rod, needle or hollow structures. The hydrothermal process has several benefits over both conventional and nonconventional methods, including cost-effective, simple, and safe for the environment. Furthermore, by using hydrothermal methods it is possible to obtain a variety of particle sizes and morphologies in aqueous solutions, which is essential for biomedical purposes. The hydrothermal technique has the ability to be hybridized with different synthesis techniques such as ultrasound, microwaves, optical radiation, hot-pressing, electrochemistry, and mechanochemistry. This combination allows for synergistic effects, leading to improved reaction kinetics, enhanced control over the synthesis process, and the development of advanced materials with tailored properties [139,140].

#### HA-Based MNCs Obtained by Hydrothermal Method

Mondal et al. (2017) synthesized hydroxyapatite coated with iron oxide nanoparticles (Fe_3_O_4_-HA), for use as a highly efficient nano-heater for hyperthermia cancer treatment. Synthesized magnetic nanoparticles exhibited superparamagnetic and hydrophilic properties, with a magnetic saturation level of 40.6 emu/g. These nanoparticles, in the absence of a magnetic field, have little or no impact on cell lines, whereas the use of hyperthermia at around 45 °C induced cell death in the cancer cells. When exposed to an A/C magnetic field, the synthesized nanoparticles had a specific energy absorption rate of 85 W/g over MG-63 osteosarcoma cells, demonstrating better heating efficiency as a nano-heater as compared to conventional HA coated with iron oxide nanoparticles. It is found that Fe_3_O_4_-HA, in combination with magnetic hyperthermia, may be a useful therapeutic tool to contrast different kinds of cancer [99].

Another work presented a novel hydrothermal technique, using α-tricalcium phosphate and magnetite nanoparticles as starting reactants. The findings showed needle-shaped HA particles with a microporous structure. Additionally, the iron oxide particles exhibited aggregation both on the surface of the composite material and within its pores. At 30 wt% or below, magnetite particle aggregates were tightly trapped in the cages of rod-shaped HA nanoparticles. This composite had both micro and submicron hole sizes (400 and 0.2 μm, respectively). It is found that if the concentration of magnetite exceeds 30 wt%, the composite material fails to fully retain the particles, rendering it unsuitable for biomedical applications. When magnetite/hydroxyapatite composites with different concentrations of magnetite were subjected to a high frequency magnetic field for 10 min, the increase in temperature within the composite was found to be dependent on the concentration of magnetite. Specifically, in the case of a composite containing 30 wt% magnetite, the temperature rise reached 55 °C, exceeding the temperature required for effective treatment. Such a composite has thus been shown to be ideal for hyperthermia therapy of bone cancer due to its ability to retain magnetite particles and generate heat [141].

HA-coated Fe_3_O_4_ nanoparticles were synthesized via an in situ hydrothermal process. It is reported that magnetic hydroxyapatite with a Fe_3_O_4_ core, synthesized using the hydrothermal method, exhibited significant superparamagnetic properties. The synthesized nanoparticles (NPs) vary in size from 19 to 24 nm and have a spherical shape with a limited size distribution that widens as the synthesis temperature rises. The saturation magnetization of core–shell nanoparticles exhibits a linear decrease as the quantity of HA increases, ranging from 68 emu/g for bare Fe_3_O_4_ core nanoparticles to 52 emu/g for Fe_3_O_4_@20HAp nanoparticles. Zeta-potential values for core–shell MNPs were measured about 30 mV, which is close to the range needed for in vivo imaging. A magnetic resonance imaging (MRI) scanning device was used to examine core–shell MNPs in an agarose matrix mimicking biological tissues. HA shell has been reported to increase the contrast in the T1-weighted mode and decrease the color’s intensity in the T2-weighted mode. The study found that there is no significant relationship between shell thickness and MRI’s contrast ability or T2/T1 intensity ratio. Therefore, the HA shell prevents the oxidation of the magnetite surface and achieves surface charge levels suitable for intravenous injection. In addition, Fe_3_O_4_@HAp core–shell MNPs have no significant effect on contrast ability, making them an intriguing candidate as a contrast agent for MRI [142].

### 2.5. Template Method

In recent years, the template method has emerged as a very interesting technique for achieving precise control in the synthesis of magnetic nanoparticles. Template synthesis method is a cutting-edge technology that has gained significant attention as a highly efficient method for synthesizing micro- and nanomaterials. This technique involves the fabrication of defined materials (usually solid compounds) inside the nanopores or channels of a nanoporous template. Template synthesis enables precise control over the shape, structure, and particle size of the resulting nanomaterials by controlling the crystal nucleation and growth during the synthesis process. The template is usually a stable and porous structure under specific reaction conditions. Within this structure, a network of nanomaterials is arranged, and the subsequent removal of the template results in the creation of a filled cavity of the desired nanomaterials bearing physical/stereochemical properties. The first step in the template synthesis involves the selection of a suitable template, which can be either a hard compound (such as polymeric microspheres, plastic foam, carbon fiber, or porous membrane) or a soft compound (such as surfactants, nanominerals, biological molecules, cells, or biopolymers) [143,144]. After the preparation of the template, the synthesis of the nanostructure of the desired material is conducted using several synthetic methods, including the sol–gel method, hydrothermal method, and co-precipitation method. Subsequently, the template is removed using etching, dissolution, and sintering, without affecting the physicochemical properties of the final product. This method has gained a lot of attention for its success in tailoring the characteristics of magnetic nanoparticles enabling them to be used as powerful tools for targeted drug delivery.

#### HA-Based MNCs Obtained by Template Method

Biomedical magnetite/hydroxyapatite nanotubes were fabricated using a natural template. Initially, tubes of magnetite (Fe_3_O_4_) were formed on a natural template, and then a bioceramic coating of hydroxyapatite was applied. A natural template refers to a porous structure that is naturally found and guides the production of magnetic hydroxyapatite, while also ensuring the proper shape and morphology due to its continuous tubular shape, narrow pore size dispersion, chemical compatibility and reproducibility. A thermal treatment at 900 °C consented to remove the natural template and improve the crystalline stability of the nanotubes. The study found that the coating with HA enhanced the biocompatibility and stability of the produced nanotubes. This HA-coated magnetite tubular structure has been found to be a viable option for the application of magnetic hyperthermia treatment when combined with an appropriate drug loading method [145].

Mir et al. (2010) [146] presented a template-assisted approach to synthesize magnetic hydroxyapatite nanocomposites (NCs), which involved the use of nano templates consisting of aqueous ferrofluids with different concentrations (20, 40, 60, and 80 μL), which were subsequently stabilized using polyvinyl alcohol. The biologically active magnetic material was produced by incorporating such polyvinyl alcohol ferrofluids (PVA-ff) onto an HA lattice structure.

The results of this study indicate that PVA-ff can be accommodated inside the HA lattice structure up to a certain point, beyond which the magnetic intra-molecular interactions become predominant, and the molecule is forced to move out from the HA structure. Nano-sized iron is hazardous to cells; thus, it must be sequestered in a non-toxic form even though it plays an essential function in living systems. To overcome this toxicity, iron nanoparticles are often coated with non-toxic, biocompatible polymers, such as PVA-ff, that creates a durable coating around the particles to stabilize the iron nanoparticles, prevent them from being toxic to cells and promote HA formation. The PVA-ff-HA nanocomposites were prepared using the wet co-precipitation process, which involves the synthesis of HA nanoparticles and protection against iron toxicity. The introduction of PVA-ff during template-assisted HA synthesis has been found to produce a unique magnetic biomaterial that has great potential as targeted delivery vehicles. These two materials have synergistic effects and are naturally biocompatible, making them a suitable platform for various biological applications [146].

Magnetic HA nanotubes were synthesized using a composite template made of polycaprolactone and magnetite NPs. Apatite minerals were deposited when the surface was activated in an alkaline pH environment. Based on morphological analysis, it was observed that a hollow tube of HA-MNPs was synthesized with HA forming an outer shell (thickness of approximately 137 nm) and most of MNPs lining the surface of the inner shell (650 nm). The magnetic hydroxyapatite nanotubes produced using this efficient method had a saturation magnetization of 27.20 emu/g, indicating a ferromagnetic character. These nanotubes have potential for biological applications, particularly in the hyperthermia therapy of bone cancer [147].

Magnetic hydroxyapatite nanoparticles were synthesized using a template of water-soluble magnetic nanoparticles composed of Fe_3_O_4_ or MnFe_3_O_4_, which were stabilized by Al(OH)_3_. Essentially, Al(OH)_3_ layers with a strongly positively charged surface and base-solubility were replaced by the neutral or slightly negative zeta potential and acid solubility of the Fe_3_O_4_-HA particles, allowing them to offer different biological properties. This synthesis method tunes the particles’ magnetic properties by changing precursor ratios without affecting radiolabeling efficiency or fluorescence properties. The synthesized Fe_3_O_4_-HA NPs have the potential to be used in biological or medical applications due to their small hydrodynamic size (50–70 nm) and high colloidal stability. These NPs have been considered as potential candidates for development as tri-modal probes for magnetic resonance imaging, positron emission tomography (PET), and optical imaging due to their fluorescent and magnetic properties, high radiolabeling efficiency for 64Cu and 18F, and excellent colloidal stability [148].

### 2.6. Sol–Gel Method

The sol–gel approach requires no specific equipment and is easy to perform. In sol–gel synthesis, reactants are converted into gels at any stage of the reaction pathway. The sol–gel technique facilitates the uniform mixing of calcium and phosphorus precursors at a relatively low processing temperature and does not require long hydrolysis periods to produce nanocrystalline powders. This method involves the production of a three-dimensional inorganic network by mixing alkoxides or other appropriate precursors in either an aqueous or organic phase, subsequently the gelation is reached by temperature increase. The gel is then dried, and any remaining organic residues are removed from the dried gel through calcination. The sol–gel technique has several benefits, including high purity, low synthesis temperature, great chemical homogeneity, and effective mixing of the initial components. Particles with excellent efficiency, purity, and control of the stoichiometry can be produced by the sol–gel method [149,150].

#### HA-Based MNCs Obtained by Sol–Gel Method

Yusoff and colleagues synthesized core–shell magnetite-hydroxyapatite nanoparticles by sol-gel method to develop an effective transport mechanism for the delivery of catechin for an improvement in nanoparticles-based drug delivery systems [151,152]. The effective electrostatic transportation of catechin by Fe_3_O_4_/HA has been attributed to the presence of the -OH component in the catechol moiety of catechin as well as the Ca^2+^ ions on the HA shell. Fe_3_O_4_ nanoparticles were spherical and capable of aggregating into larger structures. Furthermore, the synthesized nanoparticles were well dispersed and exhibited superparamagnetic ability with a saturation magnetization of 9.127 emu/g. The relationship between catechin and Fe_3_O_4_/HA has been found from the adsorption kinetics and isotherms analyses. The results indicate that the equilibrium data aligned more closely to the Langmuir isotherm model as compared to the Freundlich model. Furthermore, it was observed that the adsorption kinetics were more well described by the pseudo-second-order kinetic model as opposed to the pseudo-first-order kinetic model. This finding suggests that the primary mechanism responsible for the adsorption between catechin and Fe_3_O_4_/HA nanocomposites is electrostatic interaction. It is reported that the catechol moieties in the B ring of the catechin structure play a significant role in the chelation of Ca^2+^ ions. This adsorption study was found effective in developing a drug delivery system based on magnetic nanoparticles, especially for phytochemicals, and provided a fundamental knowledge of the adsorption mechanism [151].

Magnetic transportation of therapeutic drugs to the infection site in the human body has great potential as an effective platform for the treatment of cancer. Magnetite-hydroxyapatite (Fe_3_O_4_-HA) nanoparticles with a core–shell structure were developed by the sol–gel method to improve the safety profile, high crystalline property, and prevent the agglomeration problem. The results show that the incorporation of Fe_3_O_4_ has no effect on the phase purity or molecular structure of HA. The size of the produced Fe_3_O_4_-HA nanoparticles is around 36 nm, and they have a magnificent, monodispersed distribution. These nanoparticles have superparamagnetic properties with a saturation magnetization of 23.274 emu/g, and the EDXRF results confirmed a Ca/P ratio of 1.63, which is close to the main inorganic component of human bones. The dual affinity of magnetic Fe_3_O_4_ and highly biocompatible HA have a synergistic effect, allowing the drug or gene delivery vehicle to covertly target the infection site [152].

### 2.7. Synergistic Synthesis Method

A synergistic approach involves the integration of different synthesis methodologies, such as the combination of mechanochemical-hydro/solvothermal and microemulsion-hydro/solvothermal methods. Since this novel approach has the potential to significantly improve the properties such as size, crystallinity, morphology, and magnetic properties of the produced magnetic HA (Figure 2), it has attracted a lot of interest in nanomaterial synthesis. The synergistic strategy of combining various techniques improves their combining benefits, resulting in nanomaterials with higher crystallinity.

#### HA-Based MNCs Obtained by Synergistic Synthesis Method

A synergistic technique (chemical precipitation followed by the solvothermal process) was reported to synthesize manganese oxide-doped hydroxyapatite (Mn_3_O_4_-HA). The prepared nanocomposites, with an average size of approximately 28 nm, were found to have homogeneous spherical shape. The biocompatibility and drug loading abilities of Mn_3_O_4_-HA nanocomposites were enhanced by surface modification with the triblock copolymer Pluronic^®^ F-127 (PF-127) (FDA approved) and subsequent targeting with folic acid. The nanocomposites were developed to encapsulate metformin, with the surface properties of the polymer and the drug delivery abilities of HA. The loading capacity of the nanocomposites was found to be 98%. In an acidic environment, the synthesized nanocomposites have shown an excellent drug release profile and cell viability up to 87%. The magnetic component Mn_3_O_4_ also showed promising results in T1-Magnetic resonance imaging, with a high r_1_ relaxivity of 2.166 mM^−1^s^−1^. The produced nanocomposites were also found to have great photodynamic therapeutic efficacy under low-intensity UV irradiation, with excellent cellular uptake and production of ROS. These findings indicate that the synthesized nanocomposites have potential as targeted chemo-theranostic drugs [153].

A multifunctional cobalt ferrite/hydroxyapatite nanocomposite with a magnetic saturation value of around 2.5–8.2 emu/g were developed by an effective synergistic approach (microwave assisted and wet precipitation method). The synthesized mHA nanoparticles, followed by loading a chemotherapeutic drug (5-fluorouracil, FU) were found to function as internal heating sources when subjected to an alternating magnetic field, which increases environmental temperature, activates the polymer transition, and finally, releases a drug encapsulated in the carriers. The synthesized composites were reported to release FU at physiological temperature for up to 7 days, and their release percentage was further increased when they were accelerated at hyperthermia temperature. The synthesized magnetic composites, a thermo-responsive nano vehicle, have been shown great potential as a carrier for tumor-specific chemotherapy under hyperthermic conditions, proliferative activity against healthy fibroblast cells (L929) and suppressed growth against osteosarcoma cells (MG63) [154].

### 2.8. Microwave-Assisted Synthesis Method

Microwave-assisted synthesis works by inducing alignment of material dipoles in an external field through the excitation generated by microwave electromagnetic radiation. This method is frequently used in combination with other established synthesis strategies [155]. Microwave-assisted synthesis has been found to offer several advantages in comparison to conventional synthesis methods, including increased reaction speed, higher product yields, improved environmental sustainability, and enhanced chemical purity [156]. Microwave-assisted synthesis method has been found to be a simple, quick, cost-effective and affordable method for the efficient production of superparamagnetic HA-based materials [50].

#### 2.8.1. HA-Based MNCs Obtained by Microwave-Assisted Synthesis Method

A hierarchically nanostructured magnetic nanocomposite was formed by hybridizing Fe_3_O_4_ magnetic nanoparticles with HA ultrathin nanosheets via quick microwave-assisted method. The produced magnetic nanocomposite was characterized and investigated as a drug nanocarrier by using haemoglobin (Hb) and docetaxel (Dtxl) as model drugs. Results showed that increasing initial concentrations of Hb resulted in a higher amount of Hb adsorbed onto the magnetic nanocomposite. The release of Hb from the magnetic nanocomposite was primarily controlled by a diffusion mechanism. The magnetic nanocomposite has shown a favorable long-term release profile for Dtxl with significant pH-responsive drug release properties, which is attributed to the gradual dissolution of HA in an acidic environment. The synthesized nanocomposite was found to have a high biocompatibility, as well as a significant in vitro anticancer impact with the incorporation of Dtxl [157].

#### 2.8.2. HA Doped with Ions Giving Intrinsic Magnetic Properties Obtained by Microwave-Assisted Synthesis Method

Karunakaran et al. (2019) produced intrinsic mesoporous superparamagnetic hydroxyapatite (Fe-HA) nanoparticles by a microwave-assisted synthesis approach. Sodium dodecyl sulphate (SDS) was used as an organic modifier during the synthesis of nanoparticles. Because Fe ions have smaller ionic radii than Ca^2+^ ions, their incorporation into the HA structure is easier. The pure HA, FeHA-1, FeHA-2, and FeHA-3 nanoparticles have BET surface areas of 49, 44, 57, and 56 m^2^ g^−1^, respectively. In addition, the introduction of Fe inside the HA crystal structure led to an increase in both pore volume and pore size. The shape of magnetic nanoparticles was found to be altered by Fe incorporation, changing from a long rod-shaped morphology to a twisted form and finally to a spherical form. Thus, the incorporation of iron in HA nanoparticles resulted in the development of superparamagnetic properties. In addition, increasing the amount of Fe ions increased the saturation magnetization. The in vitro cytotoxicity analyses of both pure HA and FeHA on human osteoblast cell lines (MG63) confirmed the non-toxic behavior of NPs. Excellent antibacterial resistance was found by the antimicrobial effect of Fe-doped HA on selected microorganisms. The study found that magnetic nanoparticles could be efficiently synthesized by a SDS-assisted microwave-mediated method, enabling their potential use in many biomedical applications, such as drug targeting, hyperthermia cancer therapy, and magnetic resonance imaging [158].

Fe^3+^ and Eu^3+^ ions doped HA nanorods synthesized by microwave irradiation were found to have pure phase with dimensions of 20–30 nm in width and 50–100 nm in length. The findings show that the magnetic dipole moment produced from the unpaired electrons of the Fe^3+^ ions present in HA enables the material a paramagnetic property with simultaneous fluorescence and magnetic resonance imaging (MRI) potential in biomedicine [159].

Chandra et al. (2015) used a hydrothermal-microwave technique for the synthesis of magnetic hydroxyapatite (Co^2+^ doped HA) [160]. The results indicate that Cobalt incorporation led to a dramatic decrease in size (274–207 nm) and crystallinity, a change in morphology (from spherical to hexagonal), and an increase in magnetization (from 1.2 × 10^2^ to 6.1 × 10^2^ emug^−1^) and dielectric constant. Further, Co^2+^ ions incorporating HA exhibited enhanced haemocompatibility and sustained release of an anticancer drug (5-fluorouracil). This cobalt ion doped HA could be a promising material for magnetic imaging, drug delivery, and hyperthermia treatment.

### 2.9. Biomimetic Fabrication Methods

In recent years, the biomimetic fabrication of biological minerals—such as calcium carbonate and hydroxyapatite—has attracted a lot of interest. The biomimetic approach, which includes the use of the simulated body fluid, is the preferred method for synthesizing this composite since it offers many benefits over conventional methods (including non-toxicity and safe method) [161,162]. Conventional methods for creating calcium carbonate and hydroxyapatite, such as chemical precipitation, hydrothermal synthesis, or sol–gel processing, can be efficient, but they often come with drawbacks that restrict their use in biological applications [163,164,165]. These methods are also fast and cheap; however, they create materials with a less controlled shape and lower bioactive functions than biomimetically produced materials. Biomineralization is a dynamic process in which living organisms deposit mineral crystals within their proteinic matrix. This process leads to the formation of inorganic-based skeletal systems, such as bones or exoskeletons, which are fundamental for both structural and metabolic functions [166,167,168].

#### HA-Based MNCs Obtained by Biomimetic Fabrication Methods

Recently, Hakimi et al. (2023) evaluated the biomineralization of apatite, using a graphene oxide/HA nanocomposite. The results of the biomineralization test showed that the presence of graphene oxide combined with HA led to an increased rate of deposition of apatite-like structures. The produced nanocomposite has the potential to serve as a bioactive material for bone regeneration [169]. The incorporation of biomimetic trace elements into HA samples resulted in the formation of nanorod structures with a needle-like shape. These samples have a positively charged surface with a magnitude of 6.49 meV and showed paramagnetic properties. Ion doping during synthesis has a significant impact on these properties. The study found that introducing small amounts of multi-trace metals to HA significantly enhanced the growth of bone cells on the surface of synthesized HA. Furthermore, in vivo tests demonstrated that the magnetic HA particles were non-toxic and did not have any adverse effects on embryonic zebrafish. With these promising physical and biological properties, synthesized material has been reported as a possible biomaterial candidate for bone tissue applications [170].

The spherical–shaped mHA with a particle size of 10–200 nm was synthesized by a bio-mimetic method using simulated body fluid that included a calcium–phosphate ratio greater than 2 and a calcium–iron ratio of 0.12. Phase studies showed the presence of an iron oxide nanoparticle with a cubic spinal phase and HA crystal structures with hexagonal arrangements, but no tertiary phase was identified. The addition of a small amount of Fe has been shown to improve the mechanical properties of the scaffold, leading to enhanced cell differentiation and growth for tissue regeneration. The synthesized mHA was also found to suppress ROS formation in all concentrations due to the addition of Fe_3_O_4_ nanoparticles. Given its ability to inhibit the production of ROS species, the synthesized magnetic hydroxyapatite was also reported to have antioxidant properties, making it a valuable antioxidative material for a wide range of applications. Thus, the magnetic nanoparticles have been used in bulk for 3D printing bone defects and scaffolds, as MRI contrast reagents, and cell tracking [171].

The different synthesis methods for magnetic hydroxyapatite nanoparticles can produce particles with different morphologies shown in Figure 3. The morphology of the particles can affect their properties, such as their solubility, biocompatibility, and bioactivity [172].

Table 1 compares different methods used for synthesizing magnetic hydroxyapatite nanoparticles and their obtained properties. The parameters include the synthesized material materials, the synthesis method, particles morphology, particles size, Ca/P molar ratios, biocompatibility, and advantages. This information can be used to select the most appropriate synthesis method for a particular application.

## 3. Magnetic Materials and Stimuli in Regenerative Medicine

In the frame of Regenerative Medicine, the regeneration of bone defects, as caused by traumatic or pathologic diseases is one of the most studied approaches. Magnetic field therapies have been investigated for their potential to promote bone regeneration in various clinical cases (Figure 4). Magnetic fields can be used in combination with other treatments, such as surgery and drug delivery, to improve the outcomes of bone regeneration. Magnetic fields have been shown to induce a number of biological effects that are beneficial for bone repair, including:▪Improved mineralization: Magnetic fields stimulate the deposition of calcium and phosphate ions, thus, facilitating the mineralization process, crucial for bone formation [218,219].▪Increased cell proliferation and differentiation: Promote the growth and development of bone forming cells, known as osteoblasts into mature bone cells [220,221,222].▪Reduced inflammation: Possess the ability to reduce inflammation, which can hinder the process of bone healing [223].▪Increased angiogenesis: Promote the growth of new blood vessels, which is vital, for ensuring the supply of nutrients and oxygen to the bone [224].▪Increased antibacterial effects: Increase antibacterial effects by disrupting the bacterial cell wall, generating reactive oxygen species, and interfering with bacterial metabolism [225,226].

**Figure 4 ijms-25-02809-f004:**
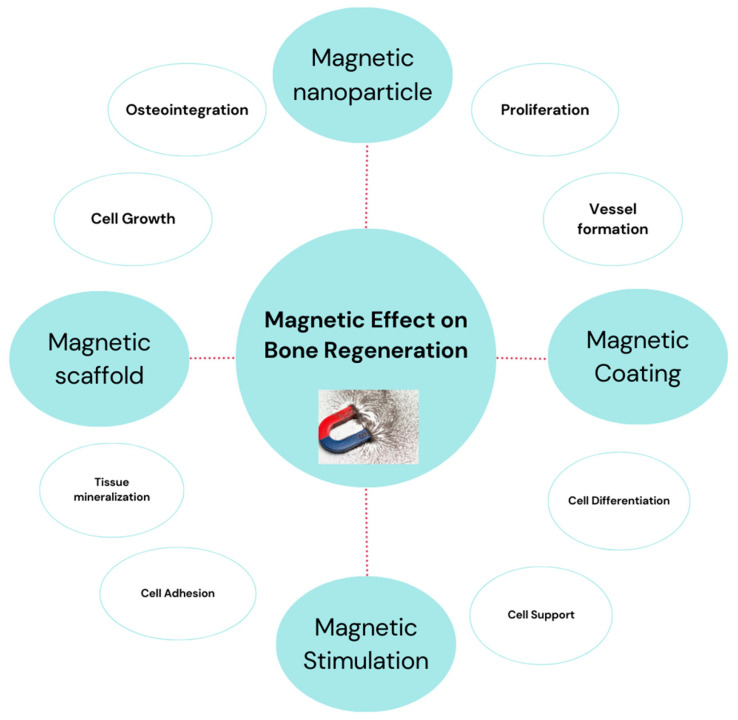
Influence of magnetic field on bone regeneration.

In addition to the intrinsic physicochemical and structural characteristics of bone scaffolds, magnetic stimulation has recently been discovered as a helpful tool in bone regeneration [100]. Faster regeneration was observed when scaffolds are used in combination with the extra physical stimulation of cells generated by external magnetic fields. Magnetic fields can be utilized in two forms: static magnetic fields (SMF) and alternating magnetic fields (AMF). It is possible to combine magnetic stimulation with magnetic materials to increase the biological response of the cells and to regulate their development and orientation in order to imitate structures with very complex architectural patterns [227].

Magnetic stimuli have a significant impact on the potential of stem cells to differentiate, with enhanced susceptibility toward the requirements of specific lineages. A recent study reported the effect of a static magnetic field of 1–2 Tesla in combination with SPIONs on the reduction of bone loss caused by unloading in vivo, as well as the formation of osteoclasts in vitro [228]. In another work, Maredziak et al. (2016) reported the enhancement of osteogenic differentiation and greater expression of related markers by human adipose-derived stem cells subjected to a SMF, whereas reducing adipogenesis suggests a specific effect of SMF towards new bone formation [229]. In another experiment, cells were magnetized by introducing magnetic nanoparticles and applying SMF; while if a graded magnetic field is generated magnetized cells can be guided towards specific targets. This research shows a correlation between differentiation and tension applied to the cells by an estimated magnetic force. The findings highlight the deformation of the nucleus as a response to mechanical stress transmitted through focal adhesions, resulting in modifications of the cytoskeleton, and subsequently, of the nuclear membrane [230].

Other authors applied magnetic fields of three different intensities to examine the effects of static magnetic fields (SMFs) on osteoblast (MC3T3-E1) responses and iron metabolism. These magnetic fields included a 500 nT hypomagnetic field (HyMF), a moderate SMF (MMF) of 0.2 T, and a high SMF (HiMF) of 16 T. After 48 h of exposure, the iron content of osteoblast cells increased in HiMF and MMF but decreased in HyMF during the growth of cells. Both the MMF and HyMF groups showed harmful effects on the differentiation of osteoblast when compared to the untreated control group that was exposed to the geomagnetic field (GMF). Specifically, both magnetic fields impeded the activity of alkaline phosphatase (ALP), as well as the processes of mineralization and calcium deposition. On the other hand, the differentiation potential exhibited an opposite trend with higher mineralization when exposed to HiMF of 16 T. During the process of cell differentiation, the iron content increased in the HyMF, remained constant in the MMF, and decreased in the HiMF. Furthermore, HyMF increased the expression of transferrin receptor 1 (TFR1) mRNA, whereas HiMF suppressed it. A higher expression level of the ferroportin 1 (FPN1) gene was observed at the same time in response to both 16 T HiMF and 0.2 T MMF [231]. Hence, osteoblast differentiation could be controlled by different SMFs with different flow intensities. Furthermore, SMFs altered the iron element during the proliferation and differentiation of osteoblasts. In this light, SMF as a non-invasive physical therapy has the potential to maintain bone health and treat bone diseases.

Comparison of studies are shown in Table 2 to explore the influence of magnetic stimulation on bone regeneration.

### 3.1. Magnetic Scaffolds

Magnetic biomaterials are considered as an intriguing candidate for bone regeneration due to their capacity for improving cellular activity and serving as a versatile therapeutic platform [219,232,233,234,235].

A variety of magnetic biomaterials are used for bone regeneration, including iron oxide nanoparticles [57], magnetic hydroxyapatite [232], magnetic composite [236], magnetically functionalized scaffolds [237], and magnetic nanoparticle-labeled cells [238], facilitating targeted therapies and improved tissue healing through magnetic field manipulation.

Magnetic scaffolds are increasingly investigated for application in bone tissue engineering, seeking to promote the bone tissue regeneration, bone healing, and regrowth process by magnetic signaling. Magnetic scaffolds have been found to enhance cell adhesion, proliferation, and differentiation [168,239,240,241,242,243,244]. Magnetic scaffolds are usually fabricated through the process of modifying or functionalizing conventional scaffold materials. Magnetic nanoparticles (MNPs) may be easily integrated into a 3D scaffold composed of several polymers by either dip-coating the scaffolds in aqueous ferrofluids containing MNPs or by dispersing the MNPs throughout the scaffold. Such features are interesting and promising for application as novel biomaterials that can be remotely controlled and directed by the application of external magnetic fields [245,246,247,248].

A novel HA-based porous ceramic composite with different ratios of magnetite (HA/Mgn 95/5, HA/Mgn 90/10, and HA/Mgn 0/50) was developed and tested in vitro with human osteoblast-like cells. The results show excellent biocompatibility, like that of a commercially available HA bone transplant, without any adverse effects, because of the presence of magnetite or the application of a static magnetic field. HA/Mgn with 90/10 ratios of magnetite has been shown to stimulate cellular growth during the early stages. Moreover, the implant has been inserted in vivo into a rabbit condyle lesion of critical size, exhibiting a significant level of histocompatibility [249]. The results suggest that magnetic HA-based scaffolds are ideal for bone tissue regeneration and provide novel ways to use magnetic fields in bone replacement, such as magnetic scaffold fixation or drug delivery.

A novel nanostructured composite scaffold was fabricated using 3D printing technology, exhibited both bone regenerative properties and magnetic characteristics, thus holding promise for applications in cancer therapy. These structures were fabricated by 3D printing a polymeric blend consisting of chitosan (CS) and polyvinyl alcohol (PVA), which incorporated hydroxyapatite nanoparticles and SPIONs. The scaffolds showed remarkable stability in a saline environment and exhibited mechanical properties comparable to the organic phase of bones. In addition, compression tests showed that introducing SPIONs to the scaffolds did not affect their mechanical properties. Based on the synergistic effects of magnetic hyperthermia treatment and increased osteogenic activity, the research found that CS/PVA/HA/SPIONs scaffolds proved to be a viable tool for bone cancer therapy and bone regeneration [250].

Another research group implanted a porous hydroxyapatite composite scaffold with superparamagnetic Fe_3_O_4_ nanoparticles into a major bone defect. A “magnetic interface” developed by fusing a magnetized scaffold to a host bone was reported to improve the bone regeneration process. The superparamagnetic nanoparticles were modified with hyperbranched poly(epsilon-Lysine) peptides and physically combined with vascular endothelial growth factor (VEGF). These nanoparticles were then administered in situ to effectively penetrate the magnetic scaffold. The findings of the study showed a significant improvement in bone regeneration at the magnetized interface. Specifically, the group treated with VEGF-MNP exhibited a higher level of regeneration compared to other groups. The nanomechanical characteristics of the tissue were comparable to the distribution of the magnetic field [251].

PCL-HA-SPION scaffolds were fabricated using 3D printing technology, incorporating varying concentrations of the superparamagnetic component. The resultant scaffolds were subsequently analyzed for their specific topographical characteristics using Atomic Force and Magnetic Force Microscopy (AFM-MFM). The homogeneity of the distribution of HA and SPION on the surface was confirmed by AFM-MFM measurements. The study showed that the 1% SPION concentration was the most effective for magnetically assisted seeding of cells in the scaffold, leading to high rates of cell entrapment and adhesion. MSCs cultivated on PCL-HA-1% SPION demonstrated excellent cell proliferation and intrinsic osteogenic potential, showing low scaffold material toxicity. According to the results, the developed PCL-HA-1% SPION scaffolds have inherent osteogenic capacity, making them excellent candidates for bone tissue repair and regeneration [252].

Xia et al. found that a calcium phosphate cement loaded with iron oxide nanoparticles (CPC-SPION) increased stem cell osteoinductivity when exposed to an external SMF [253]. This improved cellular behavior has led to a remarkable four-fold increase in bone regeneration compared to the control group treated with CPC only. The improved cellular function and bone regeneration were attributed to the physical forces produced by the magnetic field, in combination with the incorporation of magnetic nanoparticles released from the CPC-SPION constructs [235].

With the aim to increase the biomimetism of the scaffold for bone regeneration and to minimize the possible harmful effects during the scaffold degradation, which release iron oxide nanoparticles (imputed for their long-term cytotoxicity effect), recent studies focused on the development of biomaterials constituted by intrinsically magnetic apatite phases in the form of hybrid or composite scaffolds.

Concerning hybrid scaffolds containing magnetic FeHA, Tampieri et al. (2014) described a bio-inspired mineralization technique to fabricate biomimetic magnetic scaffolds. These scaffolds were obtained by mimicking the phenomena occurring in vivo during the formation of the natural bone tissue; in particular, it was induced the supramolecular assembling of Type I collagen fibrils and simultaneous nucleation of Fe^2+^/Fe^3+^-doped HA nanocrystals by pH variation at T < 40 °C, to prevent the crystallization of iron oxides. The obtained hybrid constructs showed high porosity and superparamagnetic ability. Under applied magnetic fields, the hybrid scaffolds showed enhanced cell adhesion, viability and proliferation as compared to non-magnetic scaffolds, obtained by mineralization with HA nanocrystals on self-assembling collagen [217].

The biomineralization process can also be performed by nucleating Fe/Sr co-doped HA on self-assembling collagen, providing a potential solution for bone replacement. The study showed the efficacy of ion-doped HA bio-nanomaterials in promoting osteogenic differentiation and enhancing osteoblast proliferation. The incorporation of Fe/Sr into hydroxyapatite demonstrated synergistic effects, resulting in enhanced calcium deposition, alkaline phosphatase (ALP) activity, and RUNX2 expression. The findings from the analysis of osteocalcin and osteopontin proteins in MC3T3-E1 cells provide evidence supporting the suitability and compatibility of Fe/Sr co-doped HA as a potential alternative for bone replacement [254].

In a different study, a bio-inspired mineralization process was applied to nucleate superparamagnetic FeHA nanoparticles on synthetic recombinant collagen, thus obtaining hybrid superparamagnetic FeHA-based microspheres with bone-like composition and excellent osteogenic ability with human mesenchymal stem cells. This approach permitted the obtainment of hybrid microspheres with tunable shape and size with tailored magnetization related to the amount of the inorganic FeHA component [255]. Such hybrid microspheres showed the ability to release BMP-2 growth factor with substantially bioactive effects, and to modulate the release extent by the application of pulsed electromagnetic field, thus, showing the potential of controlling the bioactivity of bio-devices by remote magnetic signal [256].

Concerning HA-based magnetic composites, a biodegradable scaffold was obtained by developing a mixture of poly(ε-caprolactone) and iron-doped FeHA nanoparticles, endowed with superparamagnetic ability. The incorporation of the apatite nanoparticles increased the hydrophilic properties of the substrate, thus, enhancing growth and viability of human mesenchymal stem cells as well as their osteogenic differentiation [257]. The use of magnetic scaffolds may represent a novel approach for boosting cell population inside the 3D scaffold structure. The in vitro tests demonstrated that the proliferation of cells in the magnetic scaffolds was 2.2-fold higher compared to the non-magnetized ones. In vivo testing in a rabbit animal model showed that the PCL/FeHA scaffolds were filled with newly formed bone over only a 4-week period, hence demonstrating excellent osteogenic properties. Incorporating magnetic properties into biocompatible materials has the potential to greatly improve 3D cell assembly, according to all of the analyses [258].

### 3.2. Magnetic Coatings

The findings evidencing the positive effects of magnetic fields applied on scaffolds to boost bone tissue regeneration have paved the way for the development of magnetic coatings, suitable to be applied on metallic implants or prosthesis to increase the bioactivity. A magnetic coating may be applied to a material in different ways. Dip coating, spin coating, and tape casting are just a few examples of cost-effective engineering processes [259,260,261,262,263]. According to some studies, the presence of magnetic particles in coating materials has been reported to facilitate the process of osteogenic proliferation in mesenchymal stem cells and adipose-derived stem cells (ASC). Furthermore, the application of an external magnetic stimulus has the potential to further enhance this effect [264,265,266,267,268,269].

Li et al. (2019) found that pulsed magnetic stimuli (PMS) on Fe_3_O_4_/mineralized collagen (FMC) coatings resulted in an improved ability of mesenchymal stem cells (MSCs) to differentiate into bone cells (osteogenic differentiation). The deformation of FMC coatings, induced by magnetic forces, was used to generate mechanical stimuli on MSCs. By manipulating the applied magnetic field and its intensity, it is easy to control the magnitude and direction of these stimuli. The study showed that MSCs cultured on FMC coatings with PMS exhibited a well-developed cytoskeleton, increased expression of integrin β1, facilitated the translocation of YAP/TAZ transcription factor into the nucleus, and improved osteogenic proliferation. The primary function of the PMS mode was to improve the osteogenic growth of MSCs by enhancing their ability to receive and transmit mechanical stimuli, hence activating the mechanical conduction mechanism by the control of the cytoskeleton movement [270]. In other work, magnetically actuated coating was developed to provide mechanical stimuli to cells to enhance osteogenic differentiation by coating deformation. The upregulated expression of associated genes suggests that the activated mechano-transduction signaling pathway was crucial for the cellular process of bone formation and differentiation [271].

The mechanism of influence of magnetic fields on bone regeneration is shown in Figure 5.

## 4. Magnetic Hydroxyapatite Nanoparticles in Nanomedicine

As already stated, the introduction of magnetic elements enables the apatite to exhibit magnetic properties; the synthesis method specifically affects the properties of the synthesized material, including its structure, crystallinity, shape, and composition. In recent times, magnetic nanoparticles (MNPs) have been used in various approaches, such as in the development of a nanostructured magnetic scaffold for tissue regeneration, hyperthermia [272], drug delivery [273] and for cell therapy [274], among others (Figure 6).

A relevant concept for advanced applications in nanomedicine is the possibility to generate magnetic cells by internalization of magnetic nanoparticles into cells themselves. Once magnetized, these cells can then be moved using magnetic forces This approach can be used for applications in bone tissue engineering, therefore, to guide cells to boost the population of implanted scaffolds or, in nanomedicine, for precisely guiding the cells to target tissues where they can deliver a therapeutic payload or exert a cell or gene therapy.

Silva et al. (2018) found that stem cells retained their functionality following magnetic targeting at a magnetic field strength ranging from 0.3 to 0.45 Tesla. However, it was observed that the proliferation rates of the stem cells decreased and there was a temporary reduction in cell viability [275]. Magnetic hydroxyapatite nanoparticles (FeHA) have a positive effect on the behavior of osteoblast-like cells, due to their intrinsic superparamagnetic properties [16]. In fact, magnetically labeled cells may be directed to the desired location using a graded magnetic field confirming they are a useful tool for cell therapy. The novel iron-doped hydroxyapatite nanoparticles (FeHA) were reported to upregulate an early marker involved in the commitment and proliferation of mesenchymal stem cells (MSCs), and to have higher cell viability as compared to SPION at four concentrations ranging from 10 μg/mL up to 200 μg/mL. Furthermore, the FeHA nanoparticles were internalized by the cells without causing any negative effects on their behavior and ultrastructure [276]. The intrinsic properties of FeHA NPs make them a promising candidate to be used as a magnetic carrier of drugs, growth factors, miRNA, etc., revealing new exciting opportunities in nanomedicine [277,278,279,280,281].

### 4.1. Magnetic Hydroxyapatite Nanoparticles as Drug Delivery Systems

Different magnetic biomaterials are used as drug delivery systems, in particular [273], magnetic hydroxyapatite [282], magnetic nanocomposite [283], magnetic microspheres [284], and magnetically responsive hydrogels [285]. These materials allow for targeted and controlled drug release to the targeted cells (like cancerous cells) by external magnetic fields (Figure 7), hence improving the therapeutic effect and reducing side effects. MNPs have ability to couple with a variety of ligands, including specific enzymes, proteins, antibodies, nucleotides, and targeted drugs. By functionalizing with these ligands, MNPs can be tailored to possess desirable properties like biocompatibility, targeting ability, and stability for in vivo applications, including but not limited to cancer therapy, drug delivery, and tissue engineering. The development of magnetic hydroxyapatite as a drug delivery platform has emerged due to its potential benefits resulting from the combination of nanoparticles’ magnetic properties with hydroxyapatite’s biocompatibility [286,287].

The controlled drug-delivery using magnetic hydroxyapatite has been affected by various crucial factors, notably particle size, surface area, and magnetic properties. These factors play a significant role in governing the precise and targeted release of therapeutic agents.

Biocompatible superparamagnetic Fe-hydroxyapatite NPs, a Helmholtz coil-based electromagnetic device, was found to be effective in stimulating the release of a model drug (ibuprofen) from FeHAs as a function of the applied frequencies, and it was integrated with a fluidic circuit simulating the flow of the cardiovascular environment. A customized magnetic/electro-magnetic bioreactor device (MEBD) typically consist of commercial generators of pulsed EMF, was developed as a potentially effective tool for drug delivery purposes, where magnetic nanoparticles (MNPs) were used to provide controlled drug release in the cardiac region without heat generation. It is found that frequencies below 100 Hz do not have an impact on the properties of cardiomyocytes (CMC), particularly contractile properties and Ca^2+^ transients [288].

Cancer therapy is a major concern in healthcare since it is the leading cause of death worldwide [289]. To design drug delivery systems for cancer therapy, particle size is believed to have a significant influence on drug solubility and bioavailability. Particularly, magnetic hydroxyapatite nanoparticles have been found to improve the bioavailability of poorly soluble drugs because of their large surface area, and this is very advantageous to face several cancer diseases [98,290,291,292].

In the treatment of cancer diseases, it has been reported that doxorubicin encapsulation efficiency may exceed 90% in the mesoporous structure of metal–polyphenol network coated magnetic hydroxyapatite nanoparticles with excellent dispersion, high specific surface area, and large pore size [293]. The desirable properties for drug loading and drug encapsulation efficiency was exhibited by magnetic hydroxyapatite nanocomposites (Fe_3_O_4_@SiO_2_@HA@ZIF-8-PEG) due to their substantial surface area [282]. Iron-doped superparamagnetic apatite (FeHA) NPs were reported to have a better affinity for the anticancer drug doxorubicin (DOX) than iron-free biomimetic HA NPs, primarily due to the strong affinity of drug for the iron cations of the FeHA surface. In fact, the amount of DOX released at pH 7.4 from FeHA was lower than the drug released from HA, indicating that the bonding stability between DOX and FeHA was higher than between DOX and HA. The evaluation of DOX release from FeHA was also examined in the presence of a pulsed electromagnetic field (PEMF), and intriguingly, the amount of DOX release after 3 and 6 days under these conditions showed a significant increase compared to the drug release in the absence of a PEMF. Since the functionalized NPs can be rapidly internalized within cells and release DOX, which accumulated in the nuclei and created cytotoxic effects, in vitro assays showed that DOX loaded on HA and FeHA was able to show its cytotoxic effects on SAOS-2 cells at the same level as free DOX, for all the concentrations and time points tested [294].

Magnetic hydroxyapatite composites in drug delivery face challenges arising from complex interaction between different phases, which have an impact on drug stability and release [105,295]. Compared to magnetic hydroxyapatite composites, single-phase magnetic hydroxyapatite is preferred for drug delivery due to its homogeneous structure, which ensures stable drug loading and release properties.

Hyperthermia is a promising treatment option in cancer therapy because of its beneficial characteristics such as its effectiveness, quick operating time, and less damage compared to chemotherapy or radiation therapy [296]. This therapeutic approach only removes tumor cells by using heat radiation, while ensuring that there are no adverse impacts on healthy microorganisms. In this approach, magnetic compounds are required to target cancer tissues by applying an external magnetic field [297]. Mondal et al. successfully proved the potential benefits of magnetic hydroxyapatite for magnetic hyperthermia cancer treatment by the fabrication of a magnetic composite with desirable characteristics. Another study has also demonstrated the potential efficacy of Magnetic Scaffolds in the context of Bone Tumor Hyperthermia Treatment [298].

### 4.2. Antimicrobial Agent

The use of different magnetic nanoparticles [299], and nanocomposites [172,300] as antimicrobial agents arises from their unique properties, which enable tailored approaches for addressing bacterial infections effectively. Magnetic hydroxyapatite is one of the promising materials for biomedical applications due to its biocompatibility, biodegradability, and magnetic properties [177,287,301,302,303]. To improve the antimicrobial properties of magnetic HA composite, low-cost and multifunctional Fe_3_O_4_@HA nanocomposites were synthesized by using natural phosphate through a simultaneous dissolution and precipitation process. The study found that when Fe_3_O_4_ was associated with apatite, it modified the surface characteristics of the Fe_3_O_4_@HA nanocomposite materials, resulting in strong antimicrobial properties against *S. aureus*, *B. subtilis*, *E. coli*, and *K. pneumoniae* [172].

To increase bacterial restriction behavior and biological properties, nanocomposites of Ce-doped HA with 90:10 (C-1), 70:30 (C-2), and 50:50 wt% (C-3) Fe_3_O_4_ NPs were produced. The synthesized Ce@HA-Fe_3_O_4_ nanocomposites were evaluated for their antibacterial efficacy against *S. aureus* and *E. coli*. Among these composites, the C-3 composite showed excellent inhibition of *E. coli* bacteria. Moreover, an in vitro biocompatibility study was conducted on the C-3 nanocomposites using MG-63 osteoblast cells based on their antibacterial activity. The results showed that the nanocomposites were highly cyto-compatible up to a concentration of 400 µg/mL, and they also showed an increase in cell viability percentage for 24 and 48 h, along with distinct cell attachment, adhesion, and development. The synthesized composites at concentrations higher than 400 µg/mL showed a slight toxic effect on the MG-63 cells, due to leaching out of Fe and Ce ions from the composite. Thus, the biological in vitro evaluations show that the synthesized nanocomposites have the potential to serve as an encouraging biomaterial for potential future uses in bone remodeling and regeneration [304].

Magnetic hyperthermia and intrinsic antimicrobial properties of magnetic hydroxyapatite have a synergistic impact, as evidenced by several recent studies proving that magnetic HA have excellent antimicrobial efficacy [92,286,305].

## 5. Conclusions and Future Perspectives

Magnetic hydroxyapatite (mHA) nanoparticles possess fascinating and unique properties, which expand their use in several applications, particularly in regenerative medicine, nanomedicine, and precision medicine. We have reviewed various synthesis methods to fabricate mHA nanoparticles including hydrothermal, chemical precipitation, mechanochemical, emulsion synthesis, sol–gel, synergistic, biomimetic, microwave-assisted, and templating methods for incorporating magnetic abilities into HA-based nanomaterials. Chemical precipitation, hydrothermal, and current template-based methods can be considered as suitable for the synthesis of mHA nanoparticles with excellent control over particle shape, size and narrow size distribution as compared to other synthesis methods. Furthermore, it is observed that the use of synergistic strategies, which include combining various synthesis techniques, can improve the control of the chemical composition, size, and shape of the mHA nanoparticles. The implementation of magnetic nanoparticles has been found to significantly improve the therapeutic ability of HA in regenerative implants, even though such toxic particles cannot be cleared by the human body neither in the long term. Targeted and effective control of drug release is made possible by using mHA nanoparticles, which can be precisely monitored and tailored under dynamic pH conditions. The superparamagnetic and biocompatible Fe-HA phase also paves the way for a new family of biomimetic, bioactive scaffolds, which have the potential to be biologically modified or activated in situ by a magnetic field, hence offering promising applications in regenerative medicine. Effective magnetic coatings for biomaterials and implants are highly demanded because of their numerous advantages, including control of the aggregation, pH-responsive assembly, enhanced bioactivity, and unique surface functions for linking various drugs and antibodies. Overall, diagnostics and therapies based on mHA nanoparticles will provide an intriguing contribution to the field of next-generation medicinal products. The success of mHA and other magnetic biomaterials will be also boosted by progress in the understanding of fundamental science related to physical, chemical, and particularly biological aspects of living tissues in interaction with magnetic fields. Several challenges still need to be addressed to improve the commercialization opportunities in the medical sector, including the rapid aggregation of FeHA nanoparticles, a deeper understanding of structure–property relation ensuring optimized performance and long-term safety, and crucial evaluations of cell-particle interactions in vitro and in vivo.

## Figures and Tables

**Figure 1 ijms-25-02809-f001:**
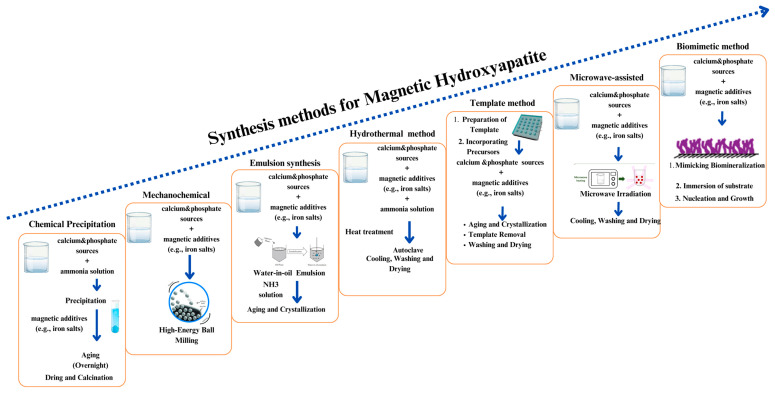
Synthesis methods for magnetic HA.

**Figure 2 ijms-25-02809-f002:**
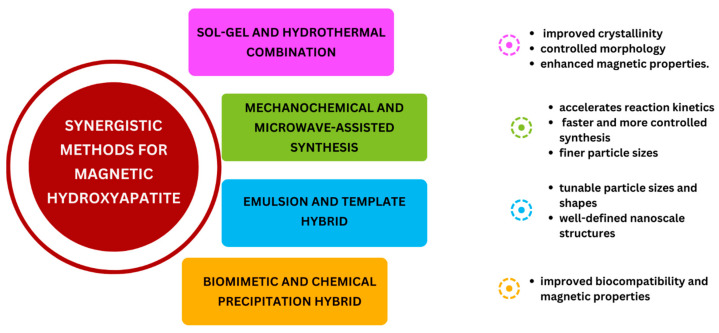
Schematic diagram of synergistic methods for magnetic hydroxyapatite.

**Figure 3 ijms-25-02809-f003:**
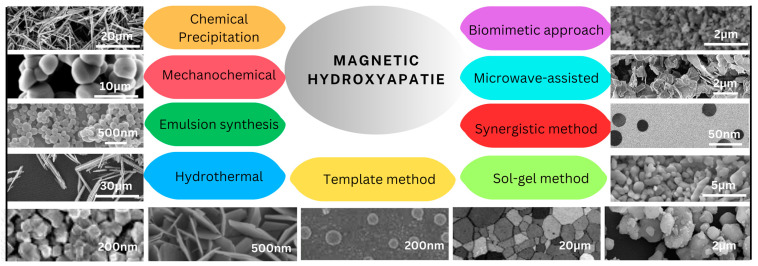
Morphologies of magnetic HA synthesized by different methods.

**Figure 5 ijms-25-02809-f005:**
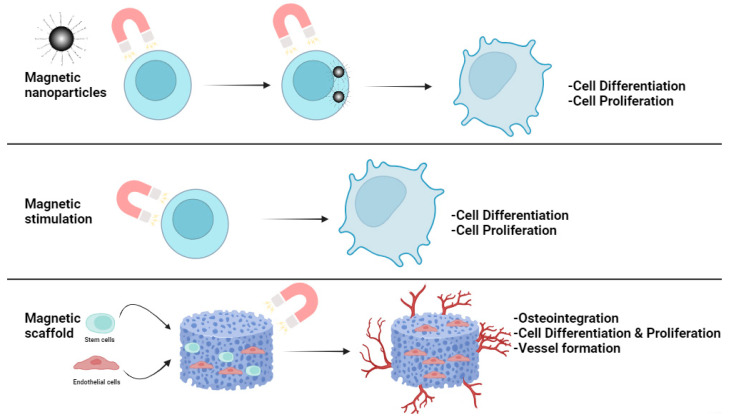
Schematic diagram of the mechanism of influence of magnetic field on bone regeneration.

**Figure 6 ijms-25-02809-f006:**
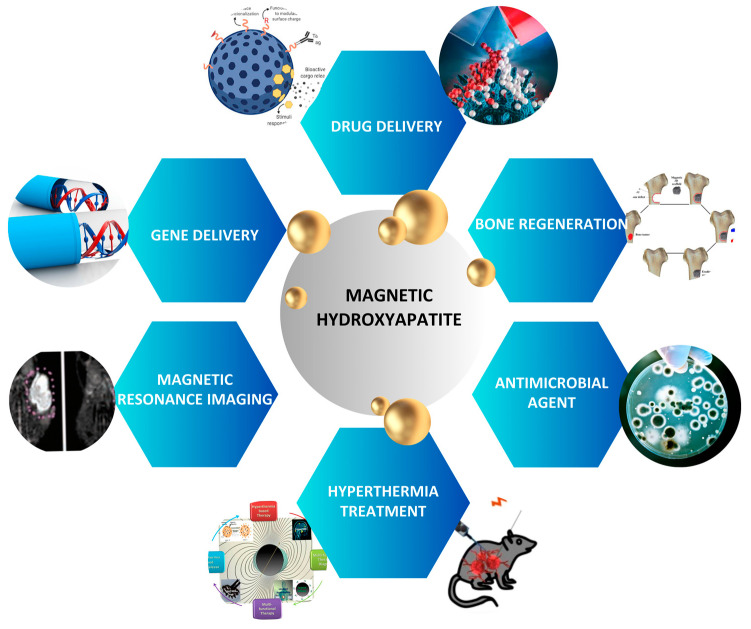
Biomedical applications of Magnetic hydroxyapatite.

**Figure 7 ijms-25-02809-f007:**
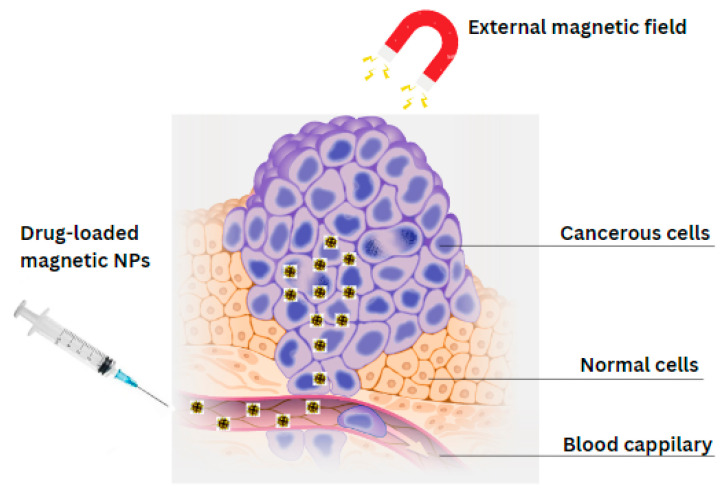
Schematic representation of controlled drug delivery by magnetic HA NPs.

**Table 1 ijms-25-02809-t001:** Comparison of different magnetic HA synthesis methods and obtained properties.

Synthesis Method	Particles Morphology	Particles Size	Ca/P Molar Ratios	Biocompatibility	Advantages	Disadvantages	Ref.
Chemical Precipitation method	Rod/Needle-like shape	10–100 nm	Limited control (1.50–1.64)	High	- eco-friendly - Low cost - Simple and quick method	Difficult to control particle morphology and size	[173,174,175,176,177,178]
Mechano-chemical method	Largely aggregated nanostructure	5–100 nm	Generally good control (1.67–1.69)	High	- Short synthesis time - Eco-friendly	- Contamination - Difficult to achieve ordered porosity, precise shape, and size due to high energy milling	[132,133,179,180,181,182]
Emulsion synthesis	Spherical shape	10–400 nm	Moderate control depending on emulsification (1.50–1.67)	Moderate	- Low production-cost - Controllable morphology - various sizes and compositions	May require the use of surfactants and stabilizers	[101,183,184,185,186,187,188]
Hydrothermal method	Rod-like shape	10–200 nm	Moderate control (1.50–1.67)	High	- Simple - Uniform and crystalline particles - Control particle size and morphology	Difficult to control particle size and morphology due to high temperature and pressure	[41,160,177,189,190,191,192,193,194,195]
Template method	Nano-spherical	5–200 nm	Good control through template selection (1.50–1.67)	High	- Controlled particle size and morphology - High surface area	- Difficulty in removing template - Risk of impurities and defects in the magnetic nanoparticles - Precise control over structure may be difficult	[184,186,196,197,198,199]
Sol-gel method	Fine-grained microstructure	10–500 nm	Good control (1.35–1.68)	High	- Low temperature processing - Ease of doping	- Long processing time - Difficult to achieve uniform particle size	[200,201,202,203,204,205]
Synergistic synthesis	Spherical	5–100 nm	Moderate control depending on synthesis methods (1.50–1.67)	High	- Efficient - Enhanced properties through synergistic effects - Controllable size limits	Complex and time-consuming	[99,153,154,206,207,208]
Microwave-assisted synthesis	Highly agglomerated hexagonal platelets and small spherical particles	10–150 nm	Moderate control depending on reaction conditions (1.50–1.67)	High	- Homogeneous product - Safe, Fast, and efficient method - Tunable magnetic properties	Non-uniform heating affect the quality and properties of synthesized nanoparticles	[178,209,210,211,212,213,214]
Biomimetic approach	Irregular, rough, and aggregated shapes	10–50 nm	Moderate control depending on biomolecules used (1.50–1.67)	High	- Mild and efficient process - Non-toxic and safe method - Bioactive properties	- Slow process - Difficult to control particle morphology and size - Use of expensive biological materials	[170,172,215,216,217]

**Table 2 ijms-25-02809-t002:** Comparison of different studies on the influence of magnetic stimulation on bone regeneration.

Type of Magnetic Field Applied	Intensity of Magnetic Field	Type of Cells	Results	Biological Effects	Advantages	Ref
HiMF of 16 T	16 T	MC3T3-E1	Iron accumulated into the cells	- Increase the level of iron in cells - Promoted osteoblast proliferation and mineralization process	- Clinical potential - used as a non-invasive physical therapy for bone health maintenance and the treatment of bone problems	[231]
1–2 T STF combined with Ferumoxytol	1–2 T	Pre-osteoclast RAW264.7 cells	- Inhibition of osteoclast formation - Reduction of Oxidative stress in osteoclast differentiation - Inhibition of the expression of NF-κB and MAPK signaling pathways	A potent tool for the translational therapy of orthopedic disorders	- Clinical potential - Prevention of damage to bone microstructure - Improvement in the mechanical properties	[228]
0.5T SMF	0.5T	Stem cells	MSCs’ osteogenic capacity is increased while their adipogenic differentiation potential is inhibited.	- A promising candidate for regulating cellular functions - improving cell proliferation	- non-toxic - Analgesic properties - anti-inflammatory properties	[229]
Static magnets	200 mT	Stem cells	Increase in osteogenesis, gene and protein expression levels	- Advancement in stem cell therapy - targeting of the implant location and cell tracking become possible	- promote osteogenic differentiation	[230]

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
