# Peer review of "Magnetic Hydroxyapatite Nanoparticles in Regenerative Medicine and Nanomedicine"

_ijms, 2024, doi:10.3390/ijms25052809_

Round 1
Reviewer 1 Report
Comments and Suggestions for Authors
Dear authors,
a very well-written review, only few remarks to address:
1) what is the purpose of dividing chapter 2 (methods) into a) and b)? Are they only reporting on different studies? Then in my opinion there is no need to divide the chapter that way. However, if you would like to point out something, then maybe use 2.1.1 headings and write a title (what are you point out?). The a), b) are a bit confusing. Please, restructure.
2) Title 4.1 should be "Magnetic hydroxyapatite nanoparticles as..." as it is describing magnetic hydroxyapatite NPs not NPs in general.
3) In "Author contributions" the part "For research articles with several authors, a short paragraph specifying their individual contributions must be provided. The following statements should be used" should be removed as they are only of orientative nature of what to put in the mentioned section.
Author Response
1) what is the purpose of dividing chapter 2 (methods) into a) and b)? Are they only reporting on different studies? Then in my opinion there is no need to divide the chapter that way. However, if you would like to point out something, then maybe use 2.1.1 headings and write a title (what are you point out?). The a), b) are a bit confusing. Please, restructure.
Thank you for your comment and suggestion. We have used appropriate headings in place of a) and b).
2) Title 4.1 should be "Magnetic hydroxyapatite nanoparticles as..." as it is describing magnetic hydroxyapatite NPs not NPs in general.
We performed the required amendment.
3) In "Author contributions" the part "For research articles with several authors, a short paragraph specifying their individual contributions must be provided. The following statements should be used" should be removed as they are only of orientative nature of what to put in the mentioned section.
We performed the required amendment.
Reviewer 2 Report
Comments and Suggestions for Authors
Present paper describes the latest advancements in magnetic hydroxyapatite nanoparticles (MHA NPs) and their potential applications in nanomedicine and regenerative medicine. Iron based MNPs for use in medicine are commonly referred to paramagnetic or superparamagnetic compounds, therefore exhibiting magnetic properties only when an external magnetic field is applied. Many studies indicate that iron oxide nanoparticles exhibit a fair biocompatibility and generally remain inert towards cells under normal conditions.
The article describes in detail the development of synthesis methods for MHA NPs from chemical precipitation and mechanochemical synthesis to recent microwave-assisted and biomimetic methods. Synergistic methods for obtaining MHA NPs are also considered. A significant part of the review is devoted to magnetic materials and in particular MHA NPs in regenerative medicine, nanomedicine, and precision medicine. Biomedical applications of magnetic biomaterials hydroxyapatite include their utilization as drug delivery systems and antimicrobial agents in different form: particular, magnetic hydroxyapatite, magnetic nanocomposite, magnetic microspheres, and magnetically responsive hydrogels. These materials allow for targeted and controlled drug release to the targeted cells by external magnetic fields, hence improving the therapeutic effect and reducing side effects. The superparamagnetic and biocompatible Fe-HA phase paves the way for a new family of biomimetic, bioactive scaffolds, which have the potential to be biologically modified or activated in situ by a magnetic field, hence offering promising applications in regenerative medicine.
The review contains almost exhaustive material on MHA NPs in recent years. Thus, in the list of references, out of 306 references, about 170 refer to publications over the last 5 years (from 2019 to 2024). Only a few references refer to work before 2010. That is, the review is relevant and timely.
The manuscript is written clearly and understandably without frills. The paper can be published in International Journal of Material Sciences in present form.
Author Response
We thank you very much for the review and appreciation.